# Occurrence, Dietary Risk Assessment and Cancer Risk Estimates of Aflatoxins and Ochratoxin A in Powdered Baby Foods Consumed in Turkey

**DOI:** 10.3390/toxins17080366

**Published:** 2025-07-25

**Authors:** Çiğdem El, Seydi Ahmet Şengül

**Affiliations:** 1Department of Child Health and Diseases, Tayfur Ata Sökmen Medicine Faculty, Hatay Mustafa Kemal University, 31060 Hatay, Turkey; cigdemel@mku.edu.tr; 2Department of Pharmacology and Toxicology, Faculty of Veterinary Medicine, Hatay Mustafa Kemal University, 31060 Hatay, Turkey

**Keywords:** aflatoxins, baby food, cancer risk, HPLC-FLD, ochratoxin A, risk assessment

## Abstract

In this study, the aim was to determine the levels of aflatoxins and ochratoxin A (OTA) in baby food consumed in Hatay using fluorescence-detector HPLC (HPLC-FLD) and to reveal the health risks that may occur in babies through consumption of these foods. To determine the dietary intake and to reveal the health risk assessment, the estimated daily intake (EDI) for all mycotoxins, the margin of exposure (MOE) for aflatoxin B_1_ (AFB_1_), aflatoxin M_1_ (AFM_1_) and OTA, the hazard index (HI) and the consumption-related hepatocellular cancer risk for AFM_1_ were calculated. It was reported that 11.5% and 8.2% of the analyzed samples exceeded the legal limit set for AFB_1_ and OTA, respectively. However, it was found that AFM_1_ concentrations in all samples did not exceed the legal limit. Based on the estimated consumption amounts of the baby foods, the HI values calculated for AFM_1_ were below 1, and the MOE values calculated for AFB_1_ and AFM_1_ were above 10.000, indicating that the consumption of baby foods does not pose a risk regarding AFB_1_ and AFM_1_ for babies. However, it was determined in all other products, except for toddler formula, that the MOE values calculated for OTA were below 10.000, indicating that their consumption may pose serious health problems in babies.

## 1. Introduction

Baby foods, known as alternatives to breastfeeding when breast milk is not available, are important food sources for babies, who constitute the most vulnerable unit of society due to reasons such as the physiological differences, limited nutritional options, low detoxification capacity and high consumption rate relative to their body weight [1,2,3,4]. For these reasons, babies are known to be more sensitive to contaminants in food compared to adults [5,6]. The rich content of baby foods in certain food ingredients that are necessary to support the physical and mental development of babies, along with their ever-increasing market share, also make babies’ exposure to toxic food contaminants highly possible [5,7]. Especially, mycotoxin contamination from milk and cereal-based ingredients in baby foods is a major concern due to its adverse health effects on babies [8]. Cereals are the food sources that can cause the highest exposure to mycotoxins [2]. Moreover, they are also an important food source for the nutrition of babies [9]. Mycotoxins are toxic secondary metabolites that can grow in foods under certain humidity and temperature conditions, primarily produced by molds such as *Aspergillus*, *Penicillium* and *Fusarium*, posing a threat in the food chain and causing various toxicological effects from allergic reactions to death in living beings [1,3,10]. Among mycotoxins, aflatoxins and ochratoxin A (OTA) are highly toxic compounds that are most frequently encountered [1,11]. While aflatoxins are produced by molds such as *Aspergillus flavus*, *Aspergillus parasiticus* and *Aspergillus nomius*, OTA is produced by *Penicillium verrucosum*, *Aspergillus ochraceous* and *Aspergillus carbonarius* [1,10,12].

Among the four main aflatoxins known as aflatoxin B_1_ (AFB_1_), aflatoxin B_2_ (AFB_2_), aflatoxin G_1_ (AFG_1_) and aflatoxin G_2_ (AFG_2_), which cause food contamination during production and storage stages, AFB_1_ is the most toxic [13]. This toxin, a cause of primary hepatocellular carcinoma, is classified as a Group 1 carcinogen by the International Agency for Research on Cancer (IARC) [5,14]. Aflatoxin M_1_ (AFM_1_), a hydroxylated metabolite of AFB_1_, can be found in high levels in milk, dairy products and milk-based foods and baby foods when farm animals consume contaminated feed [5,11,15]. Since milk and dairy products are important nutritional sources for infants due to their mineral and protein content, the presence of AFM_1_ in milk is concerning due to its high toxicity and carcinogenic properties [12]. AFM_1_, a hepatotoxic agent, is classified as a class I human carcinogen by the IARC [14]. In addition to aflatoxins, OTA is also known to be a highly toxic substance that can be found in cereal-based baby foods [10,11]. Moreover, due to its resistance to heat treatments and the procedures applied during food production stages, it is likely that residue levels in milk-based baby foods could be high [16]. The toxin, which has nephrotoxic effects, is classified as a Group 2B possible carcinogen by the IARC [14].

The type and composition of foods, along with prolonged storage under high humidity and temperature, make them a potential target for mold and mycotoxin formation [17]. As consumption of these products may lead to significant health issues, ensuring food safety in this regard is of great importance both nationally and internationally. Furthermore, exposure to mycotoxins can occur through the consumption of milk-based foods produced from contaminated feed consumed by lactating animals [18]. It is inevitable that aflatoxins, especially those formed during storage, occur in cereal-based baby foods due to their stability during industrial processing [3]. Reliable and accurate methods are needed to detect mycotoxins, which are toxic even at very low concentrations and may be present in baby foods. However, since these toxins have different structures, using similar techniques for their detection is not appropriate [18]. The development of validated methods and reliable detection systems for the analysis of toxins has recently been the focus of attention [2]. Additionally, regulations by authorities encourage the development of more sensitive and specific analytical methods for detecting toxins in foods [8]. To determine mycotoxins in baby foods, immunochemical methods such as enzyme-linked immunosorbent assay (ELISA) and radioimmunoassay (RIA), optical immune sensors such as surface plasmon resonance detection (SPR) and optical waveguide light-mode spectroscopy (OWLS) and, most commonly, various analytical methods such as thin-layer chromatography (TLC), high-performance liquid chromatography (HPLC), liquid chromatography-mass spectrometry (LC/MS) and gas chromatography (GC) are used [19]. The ELISA method, requiring fewer sample cleanup processes compared to methods such as HPLC, is considered a rapid and suitable technique for detecting aflatoxins in foods [10]. However, HPLC and LC-MS/MS methods have been reported as the best techniques among other methods for the quantification of aflatoxins and OTA in foods due to their low detection limits, high sensitivity, high recovery rates and method validation [20]. On the other hand, these methods have disadvantages such as being expensive, time-consuming, involving complicated sample processing and requiring skilled personnel. This situation significantly restricts their widespread use [21,22]. Instead of these methods, as an alternative, electrochemical biosensors have been increasingly proposed in recent years for mycotoxin detection due to their features such as high efficiency, low-cost, ease of use, simplicity and reproducibility [22,23]. Electrochemical biosensors are an analytical method used in the detection of a molecule in a sample thanks to their high sensitivity and stability due to the synergistic effects of their components [21,22]. Specifically, the detection of aflatoxins mainly based on label-free assays, utilizing either (volt)amperometric or electrochemical impedance spectroscopy (EIS)-based detection, and competitive assays, employing enzymatic and nanoparticle labels with voltammetric or photoelectrochemical detection [24].

To the best of our knowledge, no studies have revealed the occurrence of aflatoxins and OTA or estimated the risk assessment in baby foods in Hatay province, Turkey. The main purpose of this study, hence, was to determine the levels of AFB_1_, AFM_1_, total AFs and OTA in baby foods and to assess the potential health risk with respect to consumption of baby food for babies.

## 2. Results

### 2.1. Validation Results

The validation data for all baby food samples are presented in Table 1. The method was validated for the studied toxins with the following results. The method was validated in accordance with EU 2023/2782 and was based on the LOD, LOQ, R^2^, recovery (recovery value of 70–120% for aflatoxins (AFB_1_, AFB_2_, AFG_1_, AFG_2_ and AFM_1_) and OTA) and RSD parameters for toxins [25]. LOD values for AFB_1_, AFM_1_, total AFs and OTA were found to be 0.02 µg/kg, 0.00081 µg/kg, 0.03 µg/kg and 0.05 µg/kg, respectively. LOQ values for AFB_1_, AFM_1_, total AFs and OTA were determined as 0.06 µg/kg, 0.00246 µg/kg, 0.08 µg/kg and 0.17 µg/kg, respectively. LOQ values were found to be below the maximum allowed limits for AFM_1_ and OTA (0.025 µg/kg and 0.5 µg/kg, respectively) (Table 2) [26,27]. Average recoveries of the analyzed toxins spiked into baby food were found to be between 81.73% and 99.2%. The standard calibration curve for AFB_1_, AFM_1_, total AFs and OTA was found to be linear and the regression coefficients of the calibration graphs were determined as 0.9975, 0.9976, 0.9975 and 0.9981, respectively. Repeatabilities (RSD%) were found on values of 2.68%, 6.01%, 2.57% and 2.29% for AFB_1_, AFM_1_, total AFs and OTA, respectively. Method validation allowed measurement of very low levels of aflatoxins and OTA in the analyzed baby foods.

### 2.2. Occurrence AFB_1_, AFM_1_, Total AFs and OTA in Baby Food Samples Analysed

The results of the analysis of AFB_1_ in different categories of baby food samples are presented in Table 3. AFB_1_ was detected in 17 of the 25 (68%) infant formulas analyzed and 21 of the 27 follow-on formulas (77.8%), and it was determined that 8 out of 9 toddler formulas (88.9%) were below the LOD value. In addition, 4 of the infant formulas (16%), 4 of the follow-on formulas (14.8%) and all of the 9 toddler formulas were found to be below the LOQ value. AFB_1_ was detected in only 4 out of 25 infant formulas (16%), 2 out of 27 follow-on formulas (7.4%) and 1 out of 9 toddler formulas (11.1%). All mean values were detected below the LOQ value. AFB_1_ concentrations in the analyzed baby foods were determined to be in the range of <LOD-0.440 µg/kg, <LOD-0.700 µg/kg and <LOD-0.158 µg/kg for infant formula, follow-on formula and toddler formula, respectively.

The results of the AFM_1_ analysis of the baby food samples are summarized in Table 3. It was found that 32% (*n* = 8) of the infant formulas, 29.6% (*n* = 8) of the follow-on formulas and 44.4% (*n* = 4) of the toddler formulas were below the LOD value. Additionally, 20% of the infant formulas and 22.2% of both the follow-on formulas and toddler formulas were below the LOQ value. AFM_1_ was detected in only 12 (48%) of the infant formulas, 13 (48.1%) of the follow-on formulas and 3 (33.3%) of the toddler formulas. AFM_1_ levels in infant formula, follow-on formula and toddler formula samples were detected in the range of <LOD-0.01006 µg/kg, <LOD-0.01919 µg/kg and <LOD-0.02128 µg/kg, respectively, with average levels of <LOQ, 0.00279 µg/kg and 0.00314 µg/kg.

In this study, the occurrence of total AFs (AFB_1_, AFB_2_, AFG_1_ and AFG_2_) in different categories of baby food samples was investigated, and the levels in the samples are presented in Table 3. As shown in Table 3, total AFs were detected in 5 (20%) of the infant formulas, 4 (14.8%) of the follow-on formulas and only 1 (11.1%) of the toddler formulas, where legal limits have not yet been established. Total AFs in these baby food samples were found to be on average 0.143 µg/kg, 0.145 µg/kg and <LOQ levels, within the range of <LOD-1.0332 µg/kg, <LOD-1.9100 µg/kg and <LOD-0.3600 µg/kg, respectively. It was determined that 16 of the infant formulas and follow-on formulas and 7 of the toddler formulas were below the LOD, and additionally, 4 (16%), 7 (25.9%) and 1 (11.1%) of the specified baby foods were below the LOQ.

The results of the analysis of OTA in different categories of baby foods are presented in Table 3. It was found that 44% (*n* = 11) of infant formulas, 37.1% (*n* = 10) of follow-on formulas, and 44.4% (*n* = 4) of toddler formulas had levels below the LOD. Additionally, 24% of infant formulas and 11.1% of follow-on formulas and toddler formulas were found to have levels below the LOQ. The occurrence level of OTA in follow-on formulas and toddler formulas was 51.8% (14 positives out of 27 samples) and 44.4% (4 positives out of 9 samples), respectively, while 8 (32%) samples of infant formula were found to be contaminated with OTA. The concentration of OTA in the baby foods was in the range of <LOD-2.970 µg/kg, with average values in the range of <LOQ-0.236 µg/kg.

### 2.3. Health Risk Assessment Results

In this study, the risk assessment of aflatoxins and OTA resulting from the consumption of baby foods was calculated based on the average residue levels of aflatoxins and OTA detected in marketed baby foods, the average consumption amounts and the average body weights of babies in Turkey. The evaluation of aflatoxins and OTA exposure due to baby food consumption is presented in Table 4. According to the results obtained from the current study, EDI values for AFB_1_ were 0.843, 0.364 and 0.142 ng/kg b.w./day for infant formula, follow-on formula, and toddler formula, respectively. In addition, MOE values calculated using the EDI were found to be above 10.000 in all baby formulas.

EDI, MOE and HI values determined for AFM_1_ are presented in Table 4. The EDI values of AFM_1_ were calculated to be 0.032, 0.023 and 0.016 ng/kg b.w./day for infant formula, follow-on formula and toddler formula, respectively. The calculations regarding the preferred HI value to evaluate the risk associated with AFM_1_ through baby food consumption are presented in Table 4. In the current study, the HI values were determined as 0.16, 0.12 and 0.08 ng/kg b.w./day for infant formula, follow-on formula and toddler formula, respectively, and the HI values determined in all baby formulas were found to be below 1. The MOE values calculated for AFM_1_ were found to be above 10.000 in all baby formulas.

The EDI values determined for the total AFs in all baby foods are presented in Table 4. The EDI values of total AFs were determined to be 2.421, 1.210 and 0.413 ng/kg b.w./day for infant formula, follow-on formula and toddler formula, respectively. The EDI and MOE values determined for OTA are presented in Table 4. The EDI values of OTA were determined to be 2.979, 1.967 and 0.538 ng/kg b.w./day for infant formula, follow-on formula and toddler formula, respectively. It was determined that the calculated MOE values of OTA were 5.995, 9.081 and 33.176 for infant formula, follow-on formula and toddler formula, respectively, and all the values, except for toddler formula, were less than 10.000, unlike the MOE values determined for AFB_1_ and AFM_1_.

### 2.4. Carcinogenic Risk Assessment for AFM_1_

The annual probability of hepatocellular carcinoma (HCC) cases per 100.000 people, attributed to mycotoxins, was calculated for different age groups using the EDI and average cancer potential. Based on the prevalence of HBsAg_positive_ cases in the baby population reported by the World Health Organization (WHO), the annual HCC cases per 100.000 people due to mycotoxins are presented in Table 4. As shown in Table 4, the additional cancer risk due to AFM_1_ exposure from baby food consumption was found to be 0.00003, 0.00002 and 0.00002 per 100.000 population annually for infants aged 5, 9 and 12 months, respectively.

## 3. Discussion

### 3.1. Comparison of AFB_1_, AFM_1_, Total AFs and OTA Levels in the Current Study with the Results of Previous Studies

In this study, it was determined that 4 (16%) of the infant formula samples, 2 (7.4%) of the follow-on formula samples, and only 1 (11.1%) of the toddler formula samples exceeded the legal limit for AFB_1_ suggested by the EC [26] and TFC [27] (Table 2). This situation reveals the idea of routine hygiene practices and controls during the production phase in order to reduce the risk of AFB_1_ contamination in baby foods.

The occurrence and level of AFB_1_ in baby food samples are consistent with findings reported by other authors. For instance, a study by Ji et al. [9] in China reported no AFB_1_ residues in 820 cereal-based baby food samples analyzed. Additionally, there are studies where higher levels of AFB_1_ in baby food have been reported compared to the current study. For example, Mottaghianpour et al. [28] reported an average AFB_1_ level of 2.602 µg/kg, in the range of <LOQ−15.15 µg/kg, in 48 cereal-based baby food samples in Iran. The lower AFB_1_ concentrations in the baby food samples analyzed in the current study compared to other studies may be attributed to the adherence to production and hygiene standards in feed and dairy products in Turkey and the effective enforcement of established legal limits.

Since AFM_1_ is a carcinogenic substance, the potential health risks to infants resulting from exposure need to be considered [29]. For this reason, numerous studies have been conducted in Turkey and many other countries to determine the amount of AFM_1_ residue in baby food. A study by Kabak [1] investigated the AFM_1_ levels in 62 baby foods of different categories using the HPLC-FLD method. The average AFM_1_ levels in the analyzed infant formula, follow-on formula and toddler formula samples were reported as 0.016 µg/kg, 0.018 µg/kg and 0.020 µg/kg, respectively. A study by Er et al. [30] found that the average AFM_1_ levels in 84 baby food samples collected from Ankara (50 follow-on milks and 34 infant formulas) were 0.0089 µg/kg and 0.0061 µg/kg, respectively. Hooshfar et al. [31] reported an average AFM_1_ level of 0.0217 ng/g in infant formula milk samples from Iran. Omar [12] reported an average AFM_1_ level of 0.120 µg/kg, in the range of 0.0165–0.154 µg/kg, in the analyzed infant formula milk samples. The higher AFM_1_ levels in baby food samples reported in these studies compared to the current study may be due to factors such as contamination of animal feed with AFB_1_ and seasonal variations, which influence the formation of AFM_1_.

Gómez-Arranz and Navarro-Blasco [29] investigated the presence of AFM_1_ in 69 baby food samples collected in Spain. They found the average AFM_1_ level to be 0.0031 µg/kg, with a range of 0.0006–0.0115 µg/kg. Demir and Ağaoğlu [17] conducted a study in Turkey on 72 baby food samples consumed by different age groups and reported average AFM_1_ concentrations in the range of 0.0015–0.0223 µg/kg. Compared to the results of these studies, the AFM_1_ levels in the current study are found to be similar.

The levels of AFM_1_ detected in all examined baby food samples were below the legal limit of 0.025 µg/kg (Table 2) [26,27]. However, this does not imply that the consumption of baby foods available in Turkey is without risk or does not pose a threat to public health.

In a study by Sahindokuyucu Kocasari [32], AFM_1_ was not detected in any of the 41 infant formula samples analyzed. It was noted that the AFM_1_ levels detected using the ELISA method in the cited study were higher compared to those detected in the current study. These differences may arise from differing analytical methods, geographical regions, climatic factors and seasonal variability. The HPLC method is considered to have higher validity and reliability compared to the ELISA method [10].

A study by Cano-Sancho et al. [33] in Spain examined total AFs in 154 baby food samples but reported no total aflatoxin residues in any of the analyzed samples. Similarly, Razzazi-Fazeli et al. [34] reported no total AFs residues in 12 baby food samples analyzed in Indonesia. It was found that our samples were more contaminated compared to those in the mentioned studies. Since aflatoxins are considered carcinogenic and can increase the risk of hepatocellular carcinoma in infants [35], and although toxins can be present in cereal-based baby foods, there are no established residue limits for the total of AFB_1_, AFB_2_, AFG_1_ and AFG_2_ toxins in EC and TFC regulations. From a toxicological perspective, this situation poses a risk to infant health; therefore, the determination of maximum levels of these toxins in baby foods should be considered for consumption safety.

Among the tested samples, only 4 (16%) of the 25 infant formula samples and 1 (3.7%) of the 27 follow-on formula samples had OTA levels exceeding the legal limit of (0.5 µg/kg) specified by the EC [26] and TFC [27], while all 9 toddler formula samples were below the established legal limit (Table 2). Although the levels found in this study were below the permitted legal maximum levels, the potential health risks associated with the presence of OTA in cereal-based foods should not be overlooked. OTA is known to contaminate cereal-based foods and is highly nephrotoxic. It has been reported to play a role in the development of chronic kidney disease known as “Balkan Endemic Nephropathy” and in the pathogenesis of urinary tract tumors [9].

Khoshnamvand et al. [36], in a study conducted in Iran, reported that the average OTA level detected was 0.42 µg/kg as a result of an analysis of 64 cereal-based baby foods. It can be thought that the reason why the levels obtained in our current study are lower than the mentioned study is due to the minimization of the contamination level in the production of baby foods consumed in our country, thanks to developing technology. Darouj et al. [37] reported that out of 42 baby food samples analyzed (30 cereal-based baby foods and 12 fruit-based baby foods), OTA was detected in 13 (43.33%) and 4 (33.33%) of the samples, respectively. The average OTA levels were found to be 0.094 µg/kg and 0.093 µg/kg, with ranges of 0.02–0.329 µg/kg and 0.019–0.156 µg/kg, respectively. In a study by Hampikyan et al. [38] on 150 baby food samples collected in Istanbul (50 infant formulas, 50 follow-on formulas and 50 cereal-based supplementary foods), both ELISA and HPLC methods were used, revealing that 52 (35%) of the samples were contaminated with OTA, using both methods. The average OTA levels detected via ELISA in the specified baby foods were 0.043 µg/kg, 0.089 µg/kg and 0.16 µg/kg, while the average OTA levels detected via HPLC were 0.037 µg/kg, 0.082 µg/kg and 0.09 µg/kg. Yacine Ware et al. [39] reported that in a study conducted in Burkina Faso, 199 baby food samples were analyzed, with AFB_1_ detected in 167 (84%) samples and OTA in 15 (8%) samples. The average AFB_1_ level was reported as 3.8 µg/kg, with a range of 0–87.4 µg/kg, while the average OTA level was 0.1 µg/kg, with a range of 0–3.2 µg/kg. These findings are considerably higher compared to our study. This difference may be attributed to the contamination of additives used during the production of milk and cereal-based baby foods with toxins or differences in production techniques. It is known that various additives are included in baby foods to make them closer to breast milk and to support immunity. Therefore, factors such as the main ingredient (milk), production techniques, storage conditions, additives and supplements being contaminated with mycotoxins play a significant role in determining the levels of toxins observed in these foods [16].

When we look at the literature, there are few studies in Turkey and around the world revealing aflatoxins and OTA contamination in baby foods. In addition, in most studies, it was determined that AFB_1_, AFM_1_ and OTA were evaluated together in baby food samples, as in the current study. Contamination of baby foods by different mycotoxins can both contribute to their synergistic effects and cause serious adverse health effects [18]. Alvito et al. [40] in Portugal, as a result of an analysis of 27 baby food samples, stated that AFB_1_, AFM_1_ and OTA contamination was present in 15 samples. It was emphasized that 1 (7%) of the positive samples contained AFB_1_ residue, 4 (27%) contained AFM_1_ and 10 (67%) contained OTA residue. Elaridi et al. [11] in Lebanon found that AFM_1_ was in the ND−0.047 µg/kg range in 74 (88%) of 84 infant formula samples and OTA was in the range of ND−0.95 µg/kg in 80 (95%) of the samples. In these studies, the levels of AFB_1_, AFM_1_ and OTA detected in baby foods were observed to be higher than those detected in the current study. It can be said that this is due to the fact that baby foods produced in these countries are more contaminated with these toxins due to unsuitable production processes and storage conditions.

The results revealed that the analyzed baby foods were contaminated in terms of AFB_1_, AFM_1_, total AFs and OTA. This situation can cause important health problems in babies, such as immunosuppression, dysfunction in mental and cognitive development and impairment of physical development [6]. Although it is not clear whether toxins detected in baby foods exceeding the permitted legal levels will pose a risk, a risk assessment can be made to reveal what level of toxicity the residue levels may cause in babies. In addition, it was observed that the LOD and LOQ values were determined to be very low in the current study, indicating that the method used was very sensitive, and therefore, it was appropriate to perform a toxin-induced risk assessment. In addition, since the foods consumed by babies in Hatay province are also consumed in many parts of Turkey, it can be thought that it can give an idea about the aflatoxins and OTA residues found in baby foods in Turkey.

### 3.2. Comparison of the Health Risk Assessment Results of AFB_1_, AFM_1_, Total AFs and OTA of Various Studies

The risk assessment studies in the literature revealing AFB_1_, AFM_1_ and OTA exposure from baby food consumption are reported in Table 5.

The risk assessment conducted in this study determined that the MOE values for AFB_1_ and AFM_1_ in all the baby food samples were above 10.000, indicating that they do not pose any significant risk related to their consumption. In different studies, the MOE value calculated for AFM_1_ in analyzed baby foods was found to be higher than 10.000, as in our current study results, indicating that their consumption does not pose a risk to babies (Table 5) [17,31,42].

It was stated that the consumption of baby foods is safe for health if the HI value is below 1, and it was determined that the consumption of the analyzed baby foods did not pose a risk in terms of AFM_1_. Hooshfar et al. [31] reported that the HI value calculated for AFM_1_ in the analyzed baby foods was less than 1 (Table 5), as was also observed in our current study (Table 4), indicating that their consumption would not pose any health risk to babies (Table 5).

Since the MOE values calculated for OTA in infant formula and follow-on formula were below 10.000, in this study, it is reported that the consumption of these would cause high health risks in babies. In the risk assessment conducted by Kabak [1], it was reported that the EDI values determined for OTA in all baby foods were lower than the values determined in the current study. Likewise, in the study conducted by Blankson and Mill-Robertson [13] on baby foods, it was observed that the EDI values calculated for total AFs were lower than the values in the current study. It can be said that this situation is due to the high total AFs and OTA contaminations in the baby foods analyzed in the current study.

Epidemiological studies have demonstrated a significant correlation between aflatoxin exposure and the risk of hepatocellular carcinoma. The annual HCC cases per 100.000 population due to AFM_1_ exposure in different age groups have been reported in several studies (Table 5) [31,42,44]. Milićević et al. [42] reported an HCC incidence of 0.00006 per 100.000 population for the age group 12–36 months in Serbia. In the current study, it was determined that the calculated HCC risk from AFM_1_ exposure due to baby food consumption would not pose a health risk for the infant population, and the risk of liver cancer would be very low. Exposure to the specified aflatoxin through baby food consumption alone does not constitute a problem.

## 4. Conclusions

The current study provides significant insights into the contamination levels of aflatoxins and OTA in milk and cereal-based powdered baby foods consumed by infants in Hatay province. The results of the study indicate that none of the analyzed baby food samples exceeded the maximum allowable limits for AFM_1_ as set by the EU and TFC. However, samples were found to exceed the permitted legal values for AFB_1_ and OTA. Additionally, a detailed health risk assessment conducted to evaluate the potential risks from the consumption of these products revealed that there is no risk associated with AFB_1_ and AFM_1_ exposure. However, because the MOE values of all the products, except for toddler formula, are below the safe limit, it has been revealed that OTA exposure as a result of consumption may pose a serious health risk and may cause an increase in the incidence of liver cancer due to consumption in the specified age groups of babies.

The results of the study recommend several measures to reduce the residue levels of aflatoxins and OTA and to address potential health issues caused by baby food consumption. These measures should include preventing the growth of molds that produce AFB_1_ in animal feed and ensuring proper hygiene and technical knowledge among producers regarding the production, processing, storage and handling conditions of dairy or cereal-based baby foods. Given that the climate conditions in the study area are conducive to mycotoxin growth and subject to frequent variability, routine analyses should be conducted not only for aflatoxins and OTA but also for other mycotoxins such as deoxynivalenol, fumonisin and zearalenone. Furthermore, the study focused solely on the risk assessment of aflatoxins and OTA exposure from baby food consumption. It is essential to consider potential health issues arising from other foods or various toxic contaminants. Based on the results, further studies are recommended to evaluate the cumulative risks of these toxins in baby foods. A notable strength of our study is the absence of adequate studies revealing dietary and consequent cancer risk assessment due to aflatoxins and OTA exposure through baby food consumption in Turkey. Therefore, we believe that the results of our study will provide valuable insights for future research.

## 5. Materials and Methods

### 5.1. Sample Collection

In the Hatay province, 61 samples of powdered baby food from 12 different brands, with varying production dates and batch numbers, were randomly collected from different salesrooms (supermarkets and pharmacies) between 2020 and 2022. The samples included 25 infant formulas, 27 follow-on formulas and 9 toddler formulas. The collected baby foods consist of both local (*n* = 39) and imported (*n* = 22) products. After collection, the baby food samples were stored in their original pouches at −20 °C under dark and dry conditions until extraction and analysis were performed.

### 5.2. Chemicals and Reagents

All the chemicals used were of analytical reagent grade. Acetonitrile, nitric acid (65%), chloroform, n-hexane, Celite^®^ 545, sodium chloride (NaCl), potassium bromide (KBr) and methanol, all of HPLC grade, were obtained from Merck (Darmstadt, Germany). Standards for AFB_1_, AFB_2_, AFG_1_, AFG_2_, AFM_1_, OTA, Tween 20 and phosphate-buffered saline (PBS) were purchased from Sigma-Aldrich (St. Louis, MO, USA). Ultrapure deionized water, used for preparing standards in all analytical steps, was produced using the Millipore Milli-Q IQ 7000 ultrapure deionized water system (Molsheim, France). Whatman filter paper No. 4 and glass microfiber filters were obtained from Macherey-Nagel (Düren, Germany).

### 5.3. Extraction Procedures

#### 5.3.1. AFM_1_

The extraction of AFM_1_ from the collected baby food samples was carried out according to the method described by Dinçel et al. [45] with slight modifications. Briefly, 40 g of baby food was weighed and placed into a blender. To this, 10 g of Celite^®^ 545, 150 mL of chloroform and 2 mL of NaCl solution were added. After homogenizing in the blender (Waring Products Co., Torrington, CT, USA) for 3 min, the mixture was filtered through filter paper. The filtrate was dried using a rotary evaporator (IKA RV 3 V Flex, IKA, Staufen, Germany) at 50 °C with a vacuum pump (Rocker 400, Rocker, Taipei, Taiwan). After evaporation, 2 mL of methanol and 98 mL of PBS were added to the remaining residue and it was thoroughly shaken. The solvent extract was then transferred to a separation funnel. To the separation funnel, 100 mL of n-hexane was added, and the phases were allowed to separate. The resulting lower phase was passed through an AFM_1_ immunoaffinity column (Aflaprep M WIDE, R-Biopharm, Glasgow, UK) using a vacuum manifold (VISIPREP^TM^, Supelco, Bellefonte, PA, USA). Then, the column was washed with 10 mL of PBS twice. After the washing process, to obtain AFM_1_, firstly 1.25 mL of HPLC grade methanol:acetonitrile (20:30, *v*/*v*) solution and then 1.25 mL of HPLC grade ultrapure deionized water were passed through the column and into clean amber vials. The collected fraction was injected into a high-performance liquid chromatography-fluorescence detector (HPLC-FLD) (Shimadzu, Kyoto, Japan) to determine the AFM_1_ content.

#### 5.3.2. Total Aflatoxins (Total AFs)

The extraction of total AFs (AFB_1_, AFB_2_, AFG_1_ and AFG_2_) from the collected baby food samples was carried out as described in the AOAC Method 999.07 [46] with slight modifications. Briefly, for the extraction procedure, 50 g powdered baby food sample was homogenized in 300 mL methanol:ultrapure deionized water (80:20, *v*/*v*), and then, 5 g NaCl was added and extracted using a blender at high speed for 10 min. The mixture was filtered through Whatman No. 4 filter paper. The 10 mL of final extract was diluted with 60 mL of PBS. Then, an immunoaffinity column (Aflaprep, R Biopharm, Glasgow, UK) previously conditioned with 10 mL of PBS and then 60 mL of diluted filtrate was passed through it. Then, 20 mL of ultrapure deionized water was passed through the column and air was removed until the column was dry. To obtain aflatoxins, 3 mL of methanol:ultrapure deionized water (1.25:1.75, *v*/*v*) was passed through the immunoaffinity column and transferred into clean amber vials. The collected fraction was injected into the HPLC-FLD device to determine the amount of aflatoxins.

#### 5.3.3. OTA

The extraction of OTA from the collected baby food samples was carried out according to the method described in the R-Biopharm A20P14.V9 application method [47]. Briefly, 25 g of baby food and 5 g of NaCl were weighed and 100 mL of methanol:ultrapure deionized water (80:20, *v*/*v*) mixture was added and mixed in a blender at high speed for 2 min. The resulting extract was filtered through Whatman No. 4 filter paper. Then, 2 mL of the resulting filtrate was taken, and 18 mL of 0.1% Tween 20 in PBS was added and again filtered through a glass microfiber filter (Whatman GF/A, Merck, Darmstadt, Germany). Next, 10 mL volume of diluted final filtrate was passed through an immunoaffinity column (Ochraprep, R-Biopharm, Glasgow, UK) at a flow rate of 2 mL per minute. Then, the column was washed by passing 20 mL of PBS through the column at a flow rate of approximately 5 mL per minute and air was passed through the column to remove residual liquid. OTA was eluted with 1.5 mL methanol and 1.5 mL ultrapure deionized water and transferred into a clean amber vials. Finally, the collected fraction was injected into the HPLC-FLD device to determine the amount of OTA.

### 5.4. HPLC-FLD Analysis Conditions for AFB_1_, AFB_2_, AFG_1_, AFG_2_, AFM_1_ and OTA 

HPLC analyses were carried out using an LC−20 AD pump, a DGU−20A 5R degasser, a colon oven CTO−20A, a fluorescence detector model RF−10AXL and a system controller SCL−10A XP. C18 Inertsil ODS−3, 4.6 mm × 150 mm, 5 μm particle size (GL Sciences Inc., Tokyo, Japan) was used as the analytical column for all analyses. The chromatographic conditions used for each mycotoxin are given in Table 6. HPLC-FLD processes were carried out with a solvent mobile phase system consisting of water:methanol (55:45, *v*/*v*) with 35 µL nitric acid and 12 mg KBr for AFB_1_, AFB_2_, AFG_1_ and AFG_2_, water:acetonitrile:methanol (68:24:8, *v*/*v*/*v*) for AFM_1_ and acetonitrile:water:acetic acid (49.5:49.5:1, *v*/*v*/*v*) for OTA. The determination of the levels of toxin types in the analyzed baby foods was determined using the calibration curves of the standards.

### 5.5. Validation of the Analytical Method

Calibration curves were established using six different calibration standards with ranges of 0.06–8 µg/kg for AFB_1_, 0.5–20 µg/kg for total AFs and 0.1–1 µg/kg for OTA and five different calibration standards with a range of 0.0024–10 µg/kg for AFM_1_, through the dilution of stock solutions. All stock and standard solutions were stored at 4 °C in clean amber vials until analysis. The LOD and LOQ for mycotoxins were calculated based on the standard deviation of a linear response and the slope of the calibration curve. The recovery experiments were conducted by spiking the blank baby food samples with analyzed mycotoxins in 12 replicates at high and low concentration levels. The LOD and LOQ were calculated using the formula specified below.LOD=3.3 × σ/SLOQ=10 × σ/S

σ: The standard deviation of the response.S: The slope of the calibration curve.

### 5.6. Health Risk Assessment

#### 5.6.1. Estimated Daily Intake (EDI)

In order to evaluate the mycotoxin exposure from baby food consumed by babies in different age groups, estimated daily intake values were calculated based on the contamination levels obtained. Estimated daily intake was determined using the formula outlined below [48].$$\mathrm{EDI}\;(\mathrm{ng}/\mathrm{kg}\;b.w./\mathrm{day})=\frac{C_{\mathrm{mycotoxin}}\;\times \;W_{\mathrm{baby}\;\mathrm{food}}}{\mathrm{BW}\;(\mathrm{kg})}$$

C_mycotoxin_: Average mycotoxin concentration in baby foods (ng/g).W_baby food_: Average daily baby food consumption (g/day).BW: Average body weight (kg) of babies in different age groups.

In Turkey, the average body weight of a baby is considered to be 6.5 kg at 5 months old, 9 kg at 9 months and 10 kg at 12 months. Additionally, the average daily consumption of baby food for a baby is 110 g, 75 g and 50 g for an infant aged 5, 9 and 12 months, respectively [1].

#### 5.6.2. Margin of Exposure (MOE)

MOE is a method of revealing the health risks of exposure to toxins such as mycotoxins [4]. This value is used to assess the health risks associated with exposure of babies to mycotoxins through baby food consumption [48]. If the MOE value is equal to or greater than 10.000, it does not cause any health risks. It is unlikely that the exposed population will experience significant adverse effects. However, if this value is lower than 10.000, it indicates that exposure to these contaminants may pose significant health concerns to the population [4,48]. The MOE value was calculated using the formula specified below [48].$$\mathrm{MOE}=\frac{{\mathrm{BMDL}}_{10}\;(\mathrm{ng}/\mathrm{kg}\;b.w./\mathrm{day})}{\mathrm{EDI}\;(\mathrm{ng}/\mathrm{kg}\;b.w./\mathrm{day})}$$

BMDL_10_: Benchmark dose lower limit.EDI: Estimated daily intake.

To calculate the MOE, a benchmark dose lower confidence limit (BMDL_10_) of 10% is used. The BMDL_10_ is the limit, considered the lower bound of a 95% confidence interval corresponding to a 10% tumor incidence in test animals. BMDL_10_ reference doses for AFB_1_, AFM_1_ and OTA were obtained from previous studies [4,44,49,50].

#### 5.6.3. Hazard Index (HI)

To assess the potential health risks of AFM_1_ exposure from the consumption of baby food, the HI calculated using the TD_50_ of AFM_1_ (TD_50_ per body weight divided by 5000) proposed by Kuiper-Goodman [51] was used. The HI was calculated using the formula specified below [4].$$\mathrm{HI}=\frac{\mathrm{EDI}\;(\mathrm{ng}/\mathrm{kg}\;b.w./\mathrm{day})}{\mathrm{RFD}\;(\mathrm{ng}/\mathrm{kg}\;b.w./\mathrm{day})}$$

EDI: Estimated daily intake.RFD: Reference dose (0.2 ng/kg b.w./day).

A HI value above 1 indicates that consumers are at risk from a health perspective [51].

#### 5.6.4. Estimated Liver Cancer Risk Due to Consumption of Baby Foods

Consumption of baby foods contaminated with mycotoxins can lead to the triggering of liver cancer. Specifically, exposure of babies infected with hepatitis viruses to contaminated foods increases the risk of hepatocellular carcinoma [42]. The cancer risk incidence was calculated using the formula specified below [50].Cancer Risk=EDI × Average Potency$$\mathrm{Average}\;\mathrm{Potency}=[0.3\;\times \;{\mathrm{HbsAg}}_{\mathrm{positive}}]+[0.01\;\times \;{\mathrm{HbsAg}}_{\mathrm{negative}}]$$

The potential values estimated by JECFA for total AFs and OTA were reported as 0.3 (cancers per year per 100.000 population per ng/kg b.w./day) for individuals positive for hepatitis B virus surface antigen (HbsAg_positive_), while for individuals negative for hepatitis B virus surface antigen (HbsAg_negative_), the value was reported as 0.01 (cancers per year per 100.000 population per ng/kg b.w./day) [52]. However, in carcinogenicity studies for AFM_1_, it was assumed that the potency of AFM_1_ is one-tenth that of AFB_1_, even in susceptible species such as the rainbow trout and Fischer rat [53]. Additionally, hepatitis B surface antigen prevalence among children under 5 years has been reported as 0.11% in Turkey [54].

### 5.7. Statistical Analysis

All samples were analyzed in triplicate. AFB_1_, AFB_2_, AFG_1_, AFG_2_, AFM_1_ and OTA values measured in baby foods after HPLC-FLD analysis were expressed as mean ± standard deviation at the µg/kg level. In statistical analysis, standard deviation was calculated using the IBM SPSS 23.0 (IBM Corp., Armonk, NY, USA) package program.

## Figures and Tables

**Table 1 toxins-17-00366-t001:** Validation parameters of the analytical method for AFB_1_, AFM_1_, total AFs and OTA.

Mycotoxin	LOD ^a^ (µg/kg)	LOQ ^b^ (µg/kg)	*R* ^2^	Recovery (%)	RSD ^c^ (%)	Measurement Uncertainty (%)
AFB_1_	0.02	0.06	0.9975	99.2	2.68	1.83
AFM_1_	0.00081	0.00246	0.9976	81.73	6.01	0.28
Total AFs	0.03	0.08	0.9975	96.6	2.57	1.03
OTA	0.05	0.17	0.9981	94.52	2.29	2.71

^a^ LOD: Limit of detection. ^b^ LOQ: Limit of quantification. ^c^ RSD: Relative standard deviation.

**Table 2 toxins-17-00366-t002:** Baby food samples exceeding the permissible legal limits for AFB_1_, AFM_1_ and OTA according to EC and TFC.

		Exceed Legal Limit *n* (%)		
Toxin Type	EC and TFC Regulation Limit (µg/kg)	Infant Formula	Follow-On Formula	Toddler Formula
AFB_1_	0.1	4 (16)	2 (7.4)	1 (11.1)
AFM_1_	0.025	ND ^a^	ND	ND
OTA	0.5	4 (16)	1 (3.7)	ND

^a^ ND: Not detected.

**Table 3 toxins-17-00366-t003:** Concentrations of AFB_1_, AFM_1_, total AFs and OTA (µg/kg) in infant formula, follow-on formula and toddler formula.

Toxins	Samples	*n* ^a^	<LOD ^b^ *n* (%)	<LOQ ^c^ *n* (%)	Positive Samples *n* (%) ^d^	Mean ± SD *^,e^ (µg/kg)	Range ^f^ (µg/kg)
AFB_1_	Infant formula	25	17 (68)	4 (16)	4 (16)	<LOQ	<LOD-0.440
	Follow-on formula	27	21 (77.8)	4 (14.8)	2 (7.4)	<LOQ	<LOD-0.700
	Toddler formula	9	8 (88.9)	ND ^g^	1 (11.1)	<LOQ	<LOD-0.158
AFM_1_	Infant formula	25	8 (32)	5 (20)	12 (48)	<LOQ	<LOD-0.01006
	Follow-on formula	27	8 (29.6)	6 (22.2)	13 (48.1)	0.00279 ± 0.004	<LOD-0.01919
	Toddler formula	9	4 (44.4)	2 (22.2)	3 (33.3)	0.00314 ± 0.006	<LOD-0.02128
Total AFs	Infant formula	25	16 (64)	4 (16)	5 (20)	0.143 ± 0.246	<LOD-1.0332
	Follow-on formula	27	16 (59.2)	7 (25.9)	4 (14.8)	0.145 ± 0.391	<LOD-1.9100
	Toddler formula	9	7 (77.8)	1 (11.1)	1 (11.1)	<LOQ	<LOD-0.3600
OTA	Infant formula	25	11 (44)	6 (24)	8 (32)	0.176 ± 0.326	<LOD-1.143
	Follow-on formula	27	10 (37.1)	3 (11.1)	14 (51.8)	0.236 ± 0.563	<LOD-2.970
	Toddler formula	9	4 (44.4)	1 (11.1)	4 (44.4)	<LOQ	<LOD-0.405

^a^ *n*: Total number of samples analyzed. ^b^ LOD: Limit of detection. ^c^ LOQ: Limit of quantification. ^d^ *n* (%): Percent samples showing detectable concentrations (>LOD and LOQ) of aflatoxins and OTA. * The mean values are calculated using the values above the LOQ. ^e^ SD: Standard deviation. ^f^ Minimum-maximum values. ^g^ ND: Not detected.

**Table 4 toxins-17-00366-t004:** Risk assessment (HI (ng/kg b.w./day), MOE), intake (EDI (ng/kg b.w./day)) and cancer risk (CR) estimates (cancer cases/per 100.000 population/per year) of AFB_1_, AFM_1_, total AFs and OTA in baby foods.

	Mycotoxins									
		AFB_1_		AFM_1_				Total AFs	OTA	
Samples	Age	EDI ^a,b,c^	MOE ^d^	EDI	MOE	HI ^e^	CR ^f^	EDI	EDI	MOE
Infant formula	5	0.843	474.7	0.032	18,046.4	0.16	0.00003	2.421	2.979	5.995
Follow-on formula	9	0.364	1100.4	0.023	24,435	0.12	0.00002	1.210	1.967	9.081
Toddler formula	12	0.142	2825.7	0.016	36,216	0.08	0.00002	0.413	0.538	33.176

^a^ The average daily food intake indicated is 110 g, 75 g and 50 g for infants of 5, 9 and 12 months of age, respectively. ^b^ The average body weights indicated are 6.5 kg, 9 kg and 10 kg for infants of 5, 9, and 12 months of age, respectively. ^c^ EDI: Estimated daily intake. ^d^ MOE: Margin of exposure. ^e^ HI: Hazard index. ^f^ CR: Cancer risk.

**Table 5 toxins-17-00366-t005:** Summary of studies on the risk assessment and cancer risk estimates of AFB_1_, AFM_1_ and OTA in baby foods for babies.

			AFB_1_	AFM_1_				OTA	References
Country	Sample	Age	EDI	EDI	MOE	HI	CR	EDI	
	Infant formula	5		0.080					
Turkey	Follow-on formula	9		0.028				0.034	[1]
	Toddler formula	12		0.021				0.305	
China	Cereal-based infant foods	0–6						0.4	[9]
Ghana	Cereal-based foods	6–11	0.005–0.852						[13]
		12–24	0.004–0.675						
Turkey	Infant formula	0–36		0.002–0.035	11,094.5–11,805.2				[17]
Iran	Infant formula milk	0–6		0.074	7671.60	0.37	0.00010		[31]
Brazil	Infant formula	1		1.18–1.26					[41]
		6		0.59–0.63					
		12		0.39–0.42					
Serbia	Infant formula	12–36		0.029	137.7–138.9		0.00006		[42]
Brazil	Infant powdered milk	1		0.236–0.253					[43]
		6		0.117–0.127					
		12		0.078–0.084					
Iran	Infant formula milk *	0–6					0.00240–0.00060		[44]
		7–8					0.00150–0.00037		
		9–12					0.00071–0.00018		

* The values indicate domestic and imported infant formula milk, respectively.

**Table 6 toxins-17-00366-t006:** Conditions of HPLC-FLD.

Mycotoxin	Flow Rate (mL/min)	Oven Temperature (°C)	Injection Volume (µL)	Wave Length (nm)	
				λex ^a^	λem ^b^
AFM_1_	1	40	100	365	435
Total AFs	1	40	100	333	460
OTA	1	40	100	360	435

^a^ λex: Excitation. ^b^ λem: Emission.

## Data Availability

The original contributions presented in this study are included in the article. Further inquiries can be directed to the corresponding author.

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
