# Peer review of "Occurrence, Dietary Risk Assessment and Cancer Risk Estimates of Aflatoxins and Ochratoxin A in Powdered Baby Foods Consumed in Turkey"

_toxins, 2025, doi:10.3390/toxins17080366_

Round 1

Reviewer 1 Report (Previous Reviewer 3)

Comments and Suggestions for Authors

The authors have adequately answered to all questions and provided the correct changes accordingly. It is now in a format acceptable to be published.

Author Response

Author Comments: Thank you for your evaluation.

Reviewer 2 Report (Previous Reviewer 4)

Comments and Suggestions for Authors

The authors have provided an improved revised version of the manuscript titled “Occurrence, dietary risk assessment and cancer risk estimates of aflatoxins and ochratoxin A in powdered baby foods consumed in Turkey”. There are still a few points that can be further clarified / modified / improved. More specifically: 

  1. Lines 72-73, “However, since these toxins have different structures, using similar techniques for their detection is not appropriate”. The meaning of this phrase is still unclear: Haven’t all toxins been analyzed with HPLC-FLD???
  2. Lines 85-86, “However, HPLC and LC-MS/MS methods have been reported as the best techniques among other methods for the quantification of aflatoxins in foods due to their low detection limits,…”: Shouldn’t it be “for the quantification of aflatoxins and OTA in foods…”??
  3. Line 87: “On the other hand” might be better than “However”.
  4. Lines 90-91: “electrochemical biosensors have been increasingly proposed in recent years…” might be better than “electrochemical biosensors have been increasingly preferred in recent years…”
  5. Lines 110-113 and Table 1: LOD and LOQ values for AFB1, AFM1 and total AFs have changed in comparison with the previous version, while LOD and LOQ values for OTA remained the same. Please, confirm and explain.
  6. Table 4: Add “months” under “Age”
  7. Lines 403-405, “A notable strength of our study is the absence of a study revealing dietary risk assessment and cancer risk of aflatoxins and OTA exposure through baby food consumption in this region”: This might be rephrased into: “A notable strength of our study is the absence of adequate studies revealing dietary and consequent cancer risk assessment due to aflatoxins and OTA exposure through baby food consumption in Turkey”.
  8. Lines 556-558, “However, in carcinogenicity studies for AFM1, assumed that the potency of AFM1 is one-tenth that of AFB1, even in susceptible species such as the rainbow trout and Fischer rat”. The phrase seems to be incomplete and its meaning is not quite clear.

Author Response

1. Lines 72-73, “However, since these toxins have different structures, using similar techniques for their detection is not appropriate”. The meaning of this phrase is still unclear: Haven’t all toxins been analyzed with HPLC-FLD???

Author comments: In our study, aflatoxins and ochratoxin A in infant formula were analyzed by HPLC-FLD. The previous sentence contains the word mycotoxins. It is known that there are other types of mycotoxins besides aflatoxins and ochratoxin A. Different techniques are used to detect different toxins. In addition, another situation expressed here is the need for reliable and accurate methods as mentioned in the previous sentence. This means reliable validation and accurate extraction methods.

2. Lines 85-86, “However, HPLC and LC-MS/MS methods have been reported as the best techniques among other methods for the quantification of aflatoxins in foods due to their low detection limits,…”: Shouldn’t it be “for the quantification of aflatoxins and OTA in foods…”??

Author comments: Necessary corrections requested by the Reviewer were provided by the authors in the manuscript.

3. Line 87: “On the other hand” might be better than “However”.

Author comments: Necessary corrections requested by the Reviewer were provided by the authors in the manuscript.

4. Lines 90-91: “electrochemical biosensors have been increasingly proposed in recent years…” might be better than “electrochemical biosensors have been increasingly preferred in recent years…”

Author comments: Necessary corrections requested by the Reviewer were provided by the authors in the manuscript.

5. Lines 110-113 and Table 1: LOD and LOQ values for AFB1, AFM1 and total AFs have changed in comparison with the previous version, while LOD and LOQ values for OTA remained the same. Please, confirm and explain.

Author comments: As a result of the validation calculation using the new validation formula shown in the materials and methods section, it was found that the LOD and LOQ values calculated for ochratoxin A were similar to the values found in the previous article. Therefore, the same numbers were written in the article.

6. Table 4: Add “months” under “Age”

Author comments: Necessary corrections requested by the Reviewer were provided by the authors in the manuscript.

7. Lines 403-405, “A notable strength of our study is the absence of a study revealing dietary risk assessment and cancer risk of aflatoxins and OTA exposure through baby food consumption in this region”: This might be rephrased into: “A notable strength of our study is the absence of adequate studies revealing dietary and consequent cancer risk assessment due to aflatoxins and OTA exposure through baby food consumption in Turkey”.

Author comments: Necessary corrections requested by the Reviewer were provided by the authors in the manuscript.

8. Lines 556-558, “However, in carcinogenicity studies for AFM1, assumed that the potency of AFM1 is one-tenth that of AFB1, even in susceptible species such as the rainbow trout and Fischer rat”. The phrase seems to be incomplete and its meaning is not quite clear.

Author comments: The authors' point in this section is that cancer risk calculations should be based on the fact that AFM1 is 10 times less toxic than AFB1.

This manuscript is a resubmission of an earlier submission. The following is a list of the peer review reports and author responses from that submission.

Round 1

Reviewer 1 Report

Comments and Suggestions for Authors

The manuscript titled “Occurrence, Dietary Risk Assessment and Cancer Risk Estimates of Aflatoxins and Ochratoxin A in Powdered Baby Foods Consumed in Turkey” assessed the incidence of aflatoxins and ochratoxin A in powdered baby food samples collected during 2020 to 2022 in Turkey and attempted to estimate the risk of consuming these powdered baby food samples in view of their contamination levels. The topic sounds interesting and informative for food safety. However, there are some critical points to clarify.

Comments to the authors:
Introduction:

-L56-57: “AFM1, a hepatotoxic agent, is classified as a Group 2B possible carcinogen by the IARC…..” Please recheck and update this information.

Materials and methods:

-L372-379: Section on samples needs a thorough rewrite – how were the samples collected? What kind of vats/containers or packs were the samples prior to sampling and in what manner were they stored after collection? What quantity of each sample was collected per each year?

-L390-435: Section on Sample preparation of aflatoxins and ochratoxin A: please provide details of the column used and the R-Biopharm AG recommendation booklet or website.

-L476-487: For MOE calculation, please mention the value of BMDL10 to explain equation.

Results:

-Why was MOE for OTA evaluated?
- It is understandable that in some cases baby food consumption was low but there were also indications of high consumption. Authors should report exposure and risk estimations for low baby food consumption as well as for high baby food consumptions.

- Tables 1-4: Why did the authors using superscript “a”, “b”, “c”,…. ? What does it stand for?

- Table 3: The positive samples appear to refer those with concentrations above the limit of LOQ (Mean values), this should be defined to avoid ambiguity. It should be verified.

- Table 4: Why was the cancer risk assessment evaluated solely for AFM1 and not for AFB1? Please provide a rational for this decision. Together with this, the carcinogenic risk assessment for AFB1 should be described in the Section 2.4 (L168-176).

Discussion:

- L196-198: Based on this statement, in what way is your study consistent with that Ji et al., considering that AFB1 was detectable from your study, while Ji et al., did not. It should be discussed.

- L236-245: This paragraph should begin by presenting the results of this study, how many samples were contaminated with AFs especially AFB1, and then proceed to compare and discuss these findings with those of other studies.

- L281-283: What are the reasons for the relative low detection of FBs and OTA in baby foods in Turkey? It should be discussed.

- The limitation of this study including, the quantity of samples should be addressed.

Author Response

  1. Introduction:

-L56-57: “AFM1, a hepatotoxic agent, is classified as a Group 2B possible carcinogen by the IARC…..” Please recheck and update this information.

-L56-57: The above sentence has been revised as “AFM1, a hepatotoxic agent, is classified as class I human carcinogen by the IARC [14].”

L596-598: Additionally, reference 14 has been updated to “International Agency for Research on Cancer (IARC). Some Traditional Herbal Medicines, Some Mycotoxins, Naphthalene and Styrene; IARC Monograph on the Evaluation of Carcinogenic Risk to Humans; IARC Scientific Publication: Lyon, France, 2002; Volume 82.”

  1. Materials and methods:

-L372-379: Section on samples needs a thorough rewrite – how were the samples collected? What kind of vats/containers or packs were the samples prior to sampling and in what manner were they stored after collection? What quantity of each sample was collected per each year?

‘’The revision requested by the reviewer were made by the authors in the article.’’

L407-413: ‘’Samples were collected randomly. Prior to sampling, powdered infant formulas were collected in their original pouches. After collection, they were stored at -20 °C under dark and dry conditions until extraction and analysis. A total of 61 samples were collected within two years, regardless of sample quantity.’’

-L390-435: Section on Sample preparation of aflatoxins and ochratoxin A: please provide details of the column used and the R-Biopharm AG recommendation booklet or website.

L474-475: The column used was specified as “C18 Inertsil ODS-3, 4.6 mm x 150 mm, 5 μm particle size (GL Sciences Inc., Tokyo, Japan) was used as the analytical column for all analyses.”

-L476-487: For MOE calculation, please mention the value of BMDL10 to explain equation.

The revision requested by the reviewer were made by the authors in the article.

L517-530: MOE is a method of revealing the health risks of exposure to such as mycotoxins [4]. This value is used to assess the health risks associated with exposure of babies to mycotoxins through baby food consumption [45]. If the MOE value is equal to or greater than 10.000, it does not cause any health risks. It is unlikely that the exposed population will experience significant adverse effects. However, if this value is lower than 10.000, it indicates that exposure to these contaminants may pose significant health concerns to the population [4,45]. It has been revised to state that “To calculate the MOE, benchmark dose lower confidence limit (BMDL10) 10% is used. BMDL10 is the limit, considered the lower bound of a 95% confidence interval corresponding to a 10% tumor incidence in test animals. BMDL10 reference doses for AFB1, AFM1, and OTA were obtained from previous studies [4,40,46,47].”

  1. Results:

-Why was MOE for OTA evaluated?

OTA is a highly toxic type of mycotoxin. For this reason, the MOE value of OTA was calculated to determine consumption-related exposure. Additionally, the BMDL10 value for OTA is available in the literature (OTA: 17.86 ng/kg/day) [Reference: Aytekin Åžahin, G.; Aykemat, Y.; Yıldız, A.T.; Dishan, A.; İnanç, N.; Gönülalan, Z. Total aflatoxin and ochratoxin A levels, dietary exposure and cancer risk assessment in dried fruits in Türkiye. Toxicon 2024, 237, 107540.]

- It is understandable that in some cases baby food consumption was low but there were also indications of high consumption. Authors should report exposure and risk estimations for low baby food consumption as well as for high baby food consumptions.

Consumption data were stable, and the risk assessment of infant formulas consumed by different age groups in Türkiye was conducted based on another study carried out in Türkiye.

- Tables 1-4: Why did the authors using superscript “a”, “b”, “c”,…. ? What does it stand for?

“We used it to indicate and match the explanation of the abbreviations under the table.”

- Table 3: The positive samples appear to refer those with concentrations above the limit of LOQ (Mean values), this should be defined to avoid ambiguity. It should be verified.

Yes. Positive samples are those with values higher than the LOD and LOQ values. The revision requested by the reviewer has been made by the authors.

L172: d n (%): Percent samples showing detectable concentrations (>LOD and LOQ) of aflatoxins and OTA.

- Table 4: Why was the cancer risk assessment evaluated solely for AFM1 and not for AFB1? Please provide a rational for this decision.

“When we did a literature review, cancer risk assessment for AFM1 was generally presented and this allows us to compare it with other studies in the discussion.” For this reason, cancer risk assessment was performed only for AFM1.

  1. Discussion:

- L196-198: Based on this statement, in what way is your study consistent with that Ji et al., considering that AFB1 was detectable from your study, while Ji et al., did not. It should be discussed.

L229-233: "The calculated mean values for AFB1 in all 3 different types of infant formula were reported as <LOQ. These mean results are expressed as not detected. When the study by Ji et al. (2022) is examined, it is seen that AFB1 mean values are also expressed as not detected. This indicates that the levels of AFB1 are consistent with the findings reported by these authors."

- L236-245: This paragraph should begin by presenting the results of this study, how many samples were contaminated with AFs especially AFB1, and then proceed to compare and discuss these findings with those of other studies.

“It is inappropriate to give AFB1 results here as the paragraph cited by the reviewer refers to literature with total AFs”

- L281-283: What are the reasons for the relative low detection of FBs and OTA in baby foods in Turkey? It should be discussed.

“This is mentioned in the article and compared with other studies (below)”

-L234-238: It is observed that in Turkey, production and hygiene conditions in feed and dairy products are carefully followed and the established legal limits are effectively enforced.

L251-253:The contamination of the feed consumed by animals with AFB1 and various factors including seasonal variations

Reviewer 2 Report

Comments and Suggestions for Authors

Dear Authors,

major revisions needed.

1) Improve the introduction: current methods for the detection of Aflatoxin or OTA should be reported such as biosenors (doi.org/10.1016/j.microc.2023.108868)

2) title: awkward phrasing; Suggested: “Contamination and Risk Assessment of Aflatoxins and Ochratoxin A in Powdered Baby Foods Consumed in Turkey”.

3) Abstract: Language is unclear and grammatically incorrect, Needs better explanation of MOE and HI for non-experts, Clarify that OTA MOE values < 10,000 imply health risk.

4) Clarify how means are calculated when values are below LOQ/LOD.

5) Clearly define what “positive” sample means (above LOD or LOQ?).

6) MOE interpretation needs clearer explanation (especially thresholds).

7) Cancer risk values are extremely low, add context on their practical significance.

8) Provide clearer comparisons with recent studies.

9) Major Revision Improvements needed in writing, structure, and communication of scientific meaning.

Comments on the Quality of English Language

Minor English Editing

Author Response

1) Improve the introduction: current methods for the detection of Aflatoxin or OTA should be reported such as biosenors (doi.org/10.1016/j.microc.2023.108868)

L62-87: “The revision requested by the reviewer were made by the authors”

2) title: awkward phrasing; Suggested: “Contamination and Risk Assessment of Aflatoxins and Ochratoxin A in Powdered Baby Foods Consumed in Turkey”.

"The title of our study is original and effectively reflects the work."

3) Abstract: Language is unclear and grammatically incorrect, Needs better explanation of MOE and HI for non-experts, Clarify that OTA MOE values < 10,000 imply health risk.

"Language editing and proofreading of our manuscript will be requested from the Toxins journal's language editing service. The MOE value of OTA was found to be below 10000 in infant formula and follow-on formula. The potential consequences of the MOE value being below or above 10000 are specified in Section 5. Materials and Methods, subsection 5.6.2. Margin of Exposure (MOE). The revisions requested by the reviewerin this matter have been made by the authors and are indicated as follows.

L517-523: MOE is a method of revealing the health risks of exposure to such as mycotoxins [4]. This value is used to assess the health risks associated with exposure of babies to mycotoxins through baby food consumption [45]. If the MOE value is equal to or greater than 10.000, it does not cause any health risks. It is unlikely that the exposed population will experience significant adverse effects. However, if this value is lower than 10.000, it indicates that exposure to these contaminants may pose significant health concerns to the population [4,45].’

4) Clarify how means are calculated when values are below LOQ/LOD.

"When calculating mean values for aflatoxins and OTA, values below the LOQ value are considered zero (0). If the calculated average value is below the LOQ value, the average value is shown as <LOQ."

5) Clearly define what “positive” sample means (above LOD or LOQ?).

L172: Yes. Positive samples are those with values higher than the LOD and LOQ values

6) MOE interpretation needs clearer explanation (especially thresholds).

“As mentioned above, the requested clarification about the MOE was added to the article by the authors.”

“5. Materials and Methods, 5.6.2.  Indicated in Margin of Exposure (MOE).The revisions requested by the reviewer on this matter have been made by the authors and are presented as follows.” ‘MOE is a method of revealing the health risks of exposure to such as mycotoxins [4]. This value is used to assess the health risks associated with exposure of babies to mycotoxins through baby food consumption [45]. If the MOE value is equal to or greater than 10.000, it does not cause any health risks. It is unlikely that the exposed population will experience significant adverse effects. However, if this value is lower than 10.000, it indicates that exposure to these contaminants may pose significant health concerns to the population [4, 45].’

7) Cancer risk values are extremely low, add context on their practical significance.

"Similar results have been found in studies where cancer risk assessment was performed in the discussion section."

8) Provide clearer comparisons with recent studies.

"A comprehensive literature review was conducted, and our findings regarding the presence of these toxins and their risk assessment were compared with studies in the literature in the discussion section."

Reviewer 3 Report

Comments and Suggestions for Authors

This is a relevant manuscript regarding the occurrence and risk assessments of mycotoxins in baby foods. Unfortunately, major revisions need to be addressed in what regards the validation process as described in the next comments (see comment 6). 

Comment 1: For performance criteria compliance concerning method validation for mycotoxins, Regulation No 401/2006 is no longer in force and has been substituted in 2023 for Commission Implementing Regulation (EU) 2023/2782 (version in force from 2024 with amendments: https://eur-lex.europa.eu/legal-content/EN/TXT/?uri=CELEX%3A02023R2782-20240324). Please revise the validation as according to this regulation. For example, in the current regulation “Recovery: the average recovery should be between 70 and 120 %.”. Also, in the current regulation LOQ requirements are set, and should be calculated “LOQ: shall be ≤ 0,5*ML and should preferably be lower (≤ 0,2*ML).”. Performing calculations, all data is in order and complaint, but nonetheless, the text needs to be revised.

Comment 2: For calculation of validation parameters the authors mentioned Regulation 657/2002/EC (lines 451 to 452). On line 453, the authors state “The parameters examined for this purpose are linearity, limit of detection (LOD), limit of quantification (LOQ), and recovery”. The Regulation mentioned does not include LOD and LOQ, but rather CCα and CCβ. Also, LOD and LOQ are currently calculated as 3.3* σ/S and 10* σ/S, respectively (σ = the standard deviation of the response / S = the slope of the calibration curve). The calculation based in S/N is no longer used.

Comment 3: For maximum levels, Regulation (EC) No 1881/2006 is also no longer in force. It has been repealed by Commission Regulation (EU) 2023/915 (last amendment: https://eur-lex.europa.eu/legal-content/EN/TXT/?uri=CELEX%3A02023R0915-20250101). Please revise occurrence values as according to this regulation.

Comment 4: Introduction is quite poorly described considering the aim of the work, namely in what concerns references on this matter.

Comment 5: On Materials and methods, the composition of the formulas should be described since is very relevant for understanding the results obtained. For instance, what type of milk powder? Is it goat, sheep, cow? What is the percentage? Also, there are differences between baby formula bought on supermarkets vs pharmacies. Which were the ones from supermarkets, and the ones from pharmacies.

Comment 6: Calibration curve for AFB1 was done on the range of 0.2 to 8 µg/kg, and AFM1 on the range of 0.5-10 μg/kg. Being the maximum level of AFB1 0.1 µg/kg and of AFM1 of 0.025 µg/kg, how can the authors explain using the lowest calibration point higher than the maximum levels established? The maximum limit should be at least the middle point of the curve. Validation needs to be redone, since this affects the quantification of real samples, namely and especially for AFM1. The maximum values obtained for AFM1 (Table 3) were lower than 0.02, with means lower than 0.003 μg/kg, but the curve used to quantify these samples had the lowest point at 166 times higher than the mean obtained for real samples. The error of quantification is too high in all samples, making (unfortunately) all data after that unreliable.

Author Response

Comment 1: For performance criteria compliance concerning method validation for mycotoxins, Regulation No 401/2006 is no longer in force and has been substituted in 2023 for Commission Implementing Regulation (EU) 2023/2782 (version in force from 2024 with amendments: https://eur-lex.europa.eu/legal-content/EN/TXT/?uri=CELEX%3A02023R2782-20240324). Please revise the validation as according to this regulation. For example, in the current regulation “Recovery: the average recovery should be between 70 and 120 %.”. Also, in the current regulation LOQ requirements are set, and should be calculated “LOQ: shall be ≤ 0,5*ML and should preferably be lower (≤ 0,2*ML).”. Performing calculations, all data is in order and complaint, but nonetheless, the text needs to be revised.

L96-99: "The revisions requested by the reviewer were made by the authors in the article."

L611-614: The 21. reference was revised as follows: “European Commission (EC). Commission Implementing Regulation (EU) 2023/2782 of 14 December 2023 laying down the methods of sampling and analysis for the control of the levels of mycotoxins in food and repealing Regulation (EC) No 401/2006 (Text with EEA relevance). EUR-Lex. Available online: https://eur-lex.europa.eu/legal-content/EN/TXT/PDF/?uri=CELEX:02023R2782-20240324 (accessed on 10 May 2025).”

Comment 2: For calculation of validation parameters the authors mentioned Regulation 657/2002/EC (lines 451 to 452). On line 453, the authors state “The parameters examined for this purpose are linearity, limit of detection (LOD), limit of quantification (LOQ), and recovery”. The Regulation mentioned does not include LOD and LOQ, but rather CCα and CCβ. Also, LOD and LOQ are currently calculated as 3.3* σ/S and 10* σ/S, respectively (σ = the standard deviation of the response / S = the slope of the calibration curve). The calculation based in S/N is no longer used.

L491-492: "The revisions requested by the reviewer were made by the authors in the article."

LOD and LOQ calculations were performed using the equation provided below.

LOD= 3 x ?’0 LOQ= 10 x ?? ′

s0 ′ : Corrected standard deviation used in the calculation of LOD and LOQ.

Comment 3: For maximum levels, Regulation (EC) No 1881/2006 is also no longer in force. It has been repealed by Commission Regulation (EU) 2023/915 (last amendment: https://eur-lex.europa.eu/legal-content/EN/TXT/?uri=CELEX%3A02023R0915-20250101). Please revise occurrence values as according to this regulation.

"The revisions requested by the reviewer were made by the authors in the article."

L615-617: 22.  reference was revised as follows:  “European Commission (EC). COMMISSION REGULATION (EU) 2023/915 of 25 April 2023 on maximum levels for certain     contaminants in food and repealing Regulation (EC) No 1881/2006. EUR-Lex. Available online: https://eur-lex.europa.eu/legal-       content/EN/TXT/PDF/?uri=CELEX:02023R0915-20250101 (accessed on 10 May 2025).” olarak düzeltildi.

Comment 4: Introduction is quite poorly described considering the aim of the work, namely in what concerns references on this matter.

L62-92: "The revisions requested by the reviewer were made by the authors in the article."

Comment 5: On Materials and methods, the composition of the formulas should be described since is very relevant for understanding the results obtained. For instance, what type of milk powder? Is it goat, sheep, cow? What is the percentage? Also, there are differences between baby formula bought on supermarkets vs pharmacies. Which were the ones from supermarkets, and the ones from pharmacies.

Infant formulas, which are milk-based (cow’s milk) and cereal-based, were collected as composite samples. The samples were collected and analyzed based on the consumption levels and needs of different age groups, as stated in the literature. Infant formulas collected from supermarkets and pharmacies are similar.

Comment 6: Calibration curve for AFB1 was done on the range of 0.2 to 8 µg/kg, and AFM1 on the range of 0.5-10 μg/kg. Being the maximum level of AFB1 0.1 µg/kg and of AFM1 of 0.025 µg/kg, how can the authors explain using the lowest calibration point higher than the maximum levels established? The maximum limit should be at least the middle point of the curve. Validation needs to be redone, since this affects the quantification of real samples, namely and especially for AFM1. The maximum values obtained for AFM1 (Table 3) were lower than 0.02, with means lower than 0.003 μg/kg, but the curve used to quantify these samples had the lowest point at 166 times higher than the mean obtained for real samples. The error of quantification is too high in all samples, making (unfortunately) all data after that unreliable.

Upon examination of the issue reported by the reviewer, it was found that the calibration points used for AFB1 and AFM1 were inadvertently reported incorrectly. The requested revisions were made by the authors in the article.

L493-497: Calibration curves were constructed starting from LOQ values, with six calibration points for AFB1 and five calibration points for AFM1.

Reviewer 4 Report

Comments and Suggestions for Authors

The manuscript “Occurrence, Dietary Risk Assessment and Cancer Risk Estimates of Aflatoxins and Ochratoxin A in Powdered Baby Foods Consumed in Turkey” (manuscript ID: toxins-3624532) has several obscure and/or weak points, which do not allow publication in its present form.

Major Specific Comments:

Abstract: The analytical method(s) used for determining AFM1, AFB1 / total AFs (AFB1, AFB2, AFG1, AFG2), and OTA should be mentioned in the Abstract.

Abstract, lines 16-17, “However, it was determined in all other products, except for toddler formula that MOE values calculated for OTA were below 10.000, indicating that consumption of their poses a risk for OTA contamination in babies”. The meaning is not quite clear; please, rephrase.

Introduction, lines 33-35, “The nutritional importance of these foods, which have rich content to support the physical and mental development of babies, and the recent increase in market share make contamination from consumption inevitable”. The meaning is not quite clear; please, rephrase.

Results, lines 94-103: How exactly have the AFB1 levels been determined? Were they indirectly determined through determination of “total AFs” (AFB1, AFB2, AFG1, AFG2)? Please, specify.

Results, lines 115-117, “As shown in Table 3, total AFs were detected in 5 (20%) of the infant formulas, 4 (14.8%) of the follow-on formulas, and only 1 (11.1%) of the toddler formulas, where legal limits have not yet been established”: Legal limits for AFB1 (which is a part of total AFs -isn’t it?) are shown in Table 2; please, provide further literature information and/or explanatory comments to explain this issue / clarify this apparent discrepancy.

Results, 2.3 “Health Risk Assessment Results” and 2.4 “Carcinogenic Risk Assessment for AFM1”: Different parameters (EDI, MOE, HI, CR) have been calculated for AFB1, AFM1, AFs, OTA (Table 4). Please, further explain why and provide explanatory comments.

Discussion, lines 190-192, “In this study, 4 (16%) of the analyzed infant formula samples, 2 (7.4%) of the follow-on formula samples and only 1 (11.1%) of the toddler formula samples had the permission specified in the EC [19] and TFC [20] notifications for AFB1”. The meaning of this phrase is quite unclear.

Discussion, lines 201-205, “The lower AFB1 concentrations in the baby food samples analyzed in the current study compared to other studies may be attributed to the adherence to production and hygiene standards in feed and dairy products in Turkey, and the effective enforcement of established legal limits”: How many of the products analyzed were locally prepared vs. those imported to the country?? Are the former much more than the latter ones? (See also, Discussion, lines 257-260).

Discussion, lines 291-295, “It was stated that it was detected in the range of it is seen that the AFB1, AFM1 and OTA levels in the mentioned studies are higher than the levels detected in the current study, and it can be said that this is due to the fact that the contamination exposed during the production and storage stages of baby foods is higher in the samples analyzed in other studies”. The phrase is too complicated and the overall meaning is unclear; please, rephrase.

Discussion, lines 320-324, “Hooshfar et al. [24] reported that the calculated HI value was higher than the values reported in the current study and but would not pose a risk to human health. Similarly, it was reported that MOE values calculated for AFM1 in the analyzed baby foods in this study were higher than other studies and their consumption would not pose a health problem [24,25,37]”: Meaning unclear.

Discussion, lines 334-337, “Milićević et al. [37] reported in Serbia that the calculated HCC incidence of 0.00006 per 100.000 population for the 12-36 months age group was consistent with the value calculated for a similar age group in the present study”: Meaning rather unclear.

Conclusions, lines 366-367, “A notable strength of our study is the lack of previous research on aflatoxins and OTA in baby food samples from the study region, …”: What are the differences, if any, among baby foods available/consumed in Hatay region and those available/consumed in other regions of the country?? Please, specify.

Materials and Methods, 5.1. “Sample Collection”: Is the number of baby-food products analyzed enough to proceed to risk assessment studies?? Please, provide any statistical/literature support.

Materials and Methods, 5.5 “Validation of the Analytical Method”, lines 459-461, “The recovery experiments were conducted by spiking the blank baby food samples with analyzed mycotoxins in 12 replicates at high and low concentration levels”: Have the blank samples been pre-analyzed with an established analytical method? Please, provide more information.

Materials and Methods, 5.6.3 “Hazard Index (HI)”, line 495: What is the RFD value for AFM1?

Materials and Methods, 5.6.4 “Estimated Liver Cancer Risk Due to Consumption of Baby Foods”, lines 505-515, “The potential values estimated by JECFA for total AFs and OTA were reported as 0.3 (cancers per year per 100.000 population per ng/kg b.w./day) for individuals positive for hepatitis B virus surface antigen (HbsAgpositive), while for individuals negative for hepatitis B virus surface antigen (HbsAgnegative), the value was reported as 0.01 (cancers per year per 100.000 population per ng/kg b.w./day) [48]. However, in carcinogenicity studies for AFM1, assumed that the potency of AFM1 is one-tenth that of AFB1, even in susceptible species such as the rainbow trout and Fischer rat [49]. Additionally, hepatitis B surface antigen prevalence among children under 5 years has been reported as 0.11% in Turkey [50]”: The pieces of information presented are not adequately linked to each other and could not support / lead to a clear overall conclusion.

Author Response

Abstract: The analytical method(s) used for determining AFM1, AFB1 / total AFs (AFB1, AFB2, AFG1, AFG2), and OTA should be mentioned in the Abstract.

The journal’s abstract is limited to 200 words. For this reason, the analytical method could not be included.

Abstract, lines 16-17, “However, it was determined in all other products, except for toddler formula that MOE values calculated for OTA were below 10.000, indicating that consumption of their poses a risk for OTA contamination in babies”. The meaning is not quite clear; please, rephrase.

L16-18: The revisions requested by the reviewer were made by the authors in the article.

Introduction, lines 33-35, “The nutritional importance of these foods, which have rich content to support the physical and mental development of babies, and the recent increase in market share make contamination from consumption inevitable”. The meaning is not quite clear; please, rephrase.

L32-34: The revisions requested by the reviewer were made by the authors in the article.

Results, lines 94-103: How exactly have the AFB1 levels been determined? Were they indirectly determined through determination of “total AFs” (AFB1, AFB2, AFG1, AFG2)? Please, specify.

No, they are separate parameters. They were determined independently.

Results, lines 115-117, “As shown in Table 3, total AFs were detected in 5 (20%) of the infant formulas, 4 (14.8%) of the follow-on formulas, and only 1 (11.1%) of the toddler formulas, where legal limits have not yet been established”: Legal limits for AFB1 (which is a part of total AFs -isn’t it?) are shown in Table 2; please, provide further literature information and/or explanatory comments to explain this issue / clarify this apparent discrepancy.

The legal limit for AFB1 is shown in Table 2. Total AFs are expressed as the sum of AFB1 + AFB2 + AFG1 + AFG2. Upon reviewing the link below, it can be observed that the European Commission has not established a legal limit for total AFs.

https://eur-lex.europa.eu/legal-content/EN/TXT/PDF/?uri=CELEX:02023R0915-20250101

Results, 2.3 “Health Risk Assessment Results” and 2.4 “Carcinogenic Risk Assessment for AFM1”: Different parameters (EDI, MOE, HI, CR) have been calculated for AFB1, AFM1, AFs, OTA (Table 4). Please, further explain why and provide explanatory comments.

There is no specific reason for evaluating it this way. The literature review indicated that similar studies predominantly evaluated these parameters.

Discussion, lines 190-192, “In this study, 4 (16%) of the analyzed infant formula samples, 2 (7.4%) of the follow-on formula samples and only 1 (11.1%) of the toddler formula samples had the permission specified in the EC [19] and TFC [20] notifications for AFB1”. The meaning of this phrase is quite unclear.

L224-226: The revisions requested by the reviewer were made by the authors in the article.

Discussion, lines 201-205, “The lower AFB1 concentrations in the baby food samples analyzed in the current study compared to other studies may be attributed to the adherence to production and hygiene standards in feed and dairy products in Turkey, and the effective enforcement of established legal limits”: How many of the products analyzed were locally prepared vs. those imported to the country?? Are the former much more than the latter ones? (See also, Discussion, lines 257-260).

A total of 39 local and 22 imported samples were collected.

Discussion, lines 291-295, “It was stated that it was detected in the range of it is seen that the AFB1, AFM1 and OTA levels in the mentioned studies are higher than the levels detected in the current study, and it can be said that this is due to the fact that the contamination exposed during the production and storage stages of baby foods is higher in the samples analyzed in other studies”. The phrase is too complicated and the overall meaning is unclear; please, rephrase.

L324-328: The revisions requested by the reviewer were made by the authors in the article.

Discussion, lines 320-324, “Hooshfar et al. [24] reported that the calculated HI value was higher than the values reported in the current study and but would not pose a risk to human health. Similarly, it was reported that MOE values calculated for AFM1 in the analyzed baby foods in this study were higher than other studies and their consumption would not pose a health problem [24,25,37]”: Meaning unclear.

L356-358: The revisions requested by the reviewer were made by the authors in the article.

Discussion, lines 334-337, “Milićević et al. [37] reported in Serbia that the calculated HCC incidence of 0.00006 per 100.000 population for the 12-36 months age group was consistent with the value calculated for a similar age group in the present study”: Meaning rather unclear.

L370-371: The revisions requested by the reviewer were made by the authors in the article.

Conclusions, lines 366-367, “A notable strength of our study is the lack of previous research on aflatoxins and OTA in baby food samples from the study region, …”: What are the differences, if any, among baby foods available/consumed in Hatay region and those available/consumed in other regions of the country?? Please, specify.

There is no difference. Infant formulas consumed by infants in Hatay are also sold and consumed by infants in other regions of the country.

Materials and Methods, 5.1. “Sample Collection”: Is the number of baby-food products analyzed enough to proceed to risk assessment studies?? Please, provide any statistical/literature support.

The literature review revealed that the number of samples collected in our study is higher than or comparable to those in other studies (see the link below). The authors deemed the number of collected samples sufficient for risk assessment analysis.

https://www.sciencedirect.com/science/article/abs/pii/S0956713512000412?via%3Dihub

https://pmc.ncbi.nlm.nih.gov/articles/PMC6468729/

Materials and Methods, 5.5 “Validation of the Analytical Method”, lines 459-461, “The recovery experiments were conducted by spiking the blank baby food samples with analyzed mycotoxins in 12 replicates at high and low concentration levels”: Have the blank samples been pre-analyzed with an established analytical method? Please, provide more information.

No method was applied to the blank samples.

Materials and Methods, 5.6.3 “Hazard Index (HI)”, line 495: What is the RFD value for AFM1?

The revisions requested by the reviewer were made by the authors in the article.

L538: RFD: Reference dose (0.2 ng/kg b.w./day)

Materials and Methods, 5.6.4 “Estimated Liver Cancer Risk Due to Consumption of Baby Foods”, lines 505-515, “The potential values estimated by JECFA for total AFs and OTA were reported as 0.3 (cancers per year per 100.000 population per ng/kg b.w./day) for individuals positive for hepatitis B virus surface antigen (HbsAgpositive), while for individuals negative for hepatitis B virus surface antigen (HbsAgnegative), the value was reported as 0.01 (cancers per year per 100.000 population per ng/kg b.w./day) [48]. However, in carcinogenicity studies for AFM1, assumed that the potency of AFM1 is one-tenth that of AFB1, even in susceptible species such as the rainbow trout and Fischer rat [49]. Additionally, hepatitis B surface antigen prevalence among children under 5 years has been reported as 0.11% in Turkey [50]”: The pieces of information presented are not adequately linked to each other and could not support / lead to a clear overall conclusion.

The data required for cancer risk assessment are provided here.

Round 2

Reviewer 1 Report

Comments and Suggestions for Authors

The revised manuscript has been improved. 

Author Response

All revisions to the article were edited by the authors. Thank you.

Reviewer 2 Report

Comments and Suggestions for Authors

The authors didn't mention electrochemical biosensors as reliable electrochemical methods for AFB1 and Ochratoxin detection. However, there are several examples reported in the literature; here some examples: doi.org/10.1016/j.microc.2023.108868.

Please improve this section

Author Response

The authors didn't mention electrochemical biosensors as reliable electrochemical methods for AFB1 and Ochratoxin detection. However, there are several examples reported in the literature; here some examples: doi.org/10.1016/j.microc.2023.108868.

Please improve this section

Authors’ response: The revisions requested by the reviewer were made by the authors in the article. The authors mentioned electrochemical biosensors used for detection of mycotoxins.

L87-98: However, these methods have disadvantages such as being expensive, time-consuming, involving complicated sample processing, and requiring skilled personnel. This situation significantly restricts their widespread use [21,22]. Instead of these methods, as an alternative, electrochemical biosensors have been increasingly preferred in recent years for mycotoxin detection due to their features such as high efficiency, low-cost, ease of use, simplicity, and reproducibility [22,23]. Electrochemical biosensors are an analytical method used in the detection of a molecule in a sample thanks to its high sensitivity and stability due to the synergistic effects of its components [21,22]. Specifically, the detection of aflatoxins mainly based on label-free assays, utilizing either (volt)amperometric or electrochemical impedance spectroscopy (EIS)-based detection, and competitive assays, employing enzymatic and nanoparticle labels with voltammetric or photoelectrochemical detection [24].

Reviewer 3 Report

Comments and Suggestions for Authors

Although most changes were made as according to the first revision of the article, regarding comment 2. calculation of LOD and LOQ, authors answered with "The revisions requested by the reviewer were made by the authors in the article.". Looking at the values presented in the revised document, these are the same as the ones in the first submmited version. Being two completely different formulas (one based on signal to noise and other being based on the calibration curve), how can a value like the one from AFM1 (0.00072) be exactly the same?

Author Response

Although most changes were made as according to the first revision of the article, regarding comment 2. calculation of LOD and LOQ, authors answered with "The revisions requested by the reviewer were made by the authors in the article.". Looking at the values presented in the revised document, these are the same as the ones in the first submmited version. Being two completely different formulas (one based on signal to noise and other being based on the calibration curve), how can a value like the one from AFM1 (0.00072) be exactly the same?

Authors’ response: To determine LOD and LOQ values, blank samples were read on HPLC in 12 replicates. 3 times the standard deviation of the values ​​read was determined as LOD, 10 times as LOQ. Then, the calibration curve was drawn with the determined values ​​and the samples were read on the device.

Reviewer 4 Report

Comments and Suggestions for Authors

The authors have provided an improved revised version of the manuscript titled “Occurrence, dietary risk assessment and cancer risk estimates of aflatoxins and ochratoxin A in powdered baby foods consumed in Turkey”. Nevertheless, at least in my opinion, there are still points that can be further clarified / modified / improved. More specifically: 

1.Authors’ response: The journal’s abstract is limited to 200 words. For this reason, the analytical method could not be included.

Comment on the Authors’ response: To help the authors keep the limit of 200 words, I would propose the following slight modification to the original abstract:

“In this study, it was aimed to determine the levels of aflatoxins and ochratoxin A (OTA) in baby food consumed in Hatay using fluorescence-detector HPLC (HPLC-FLD) and to reveal the health risks that through consumption of these foods may be occur in babies. To determine dietary intake and to reveal the health risk assessment, estimated daily intake (EDI) for all mycotoxins, margin of exposure (MOE) for aflatoxin B1 (AFB1), aflatoxin M1 (AFM1), and OTA, hazard index (HI) and consumption-related hepatocellular cancer risk for AFM1 were calculated. It was reported that (11.5%) and (8.2%) of analyzed samples exceeded the legal limit set for AFB1 and OTA, respectively. However, it was found that AFM1 concentrations in all samples did not exceed the legal limit. Based on the estimated consumption amounts of the baby foods, HI values calculated for AFM1 were below 1, and MOE values calculated for AFB1 and AFM1 were above 10.000, indicating that consumption of baby foods does not pose a risk regarding AFB1 and AFM1 for babies. However, it was determined in all other products, except for toddler formula that MOE values calculated for OTA were below 10.000, indicating that their consumption may pose serious health problems in babies”. Word count: 200

2.Authors’ response: L16-18: The revisions requested by the reviewer were made by the authors in the article.

Comment on the Authors’ response: Ok

3.Authors’ response: L32-34: The revisions requested by the reviewer were made by the authors in the article.

Comment on the Authors’ response: For better comprehension, I would propose rephrasing, e.g. as follows: “The rich content of baby foods in certain food ingredients that are necessary to support the physical and mental development of babies, along with their ever-increasing market-share, also make babies' exposure to toxic food-contaminants highly possible”.

4.Authors’ response: No, they are separate parameters. They were determined independently.

Comment on the Authors’ response: Ok, thanks for clarifying.

5.Authors’ response: The legal limit for AFB1 is shown in Table 2. Total AFs are expressed as the sum of AFB1 + AFB2 + AFG1 + AFG2. Upon reviewing the link below, it can be observed that the European Commission has not established a legal limit for total AFs.

https://eur-lex.europa.eu/legal-content/EN/TXT/PDF/?uri=CELEX:02023R0915-20250101

Comment on the Authors’ response: Ok

6.Authors’ response: There is no specific reason for evaluating it this way. The literature review indicated that similar studies predominantly evaluated these parameters.

Comment on the Authors’ response: Ok

7.Authors’ response: L224-226: The revisions requested by the reviewer were made by the authors in the article.

Comment on the Authors’ response: Ok

8.Authors’ response: A total of 39 local and 22 imported samples were collected.

Comments on the Authors’ response: 1. Please, add this information to “5.1. Sample Collection”. 2. According to the authors (Discussion, lines 234-237): “The lower AFB1 concentrations in the baby food samples analyzed in the current study compared to other studies may be attributed to the adherence to production and hygiene standards in feed and dairy products in Turkey, and the effective enforcement of established legal limits”. However, a substantial part of the baby foods analyzed in this study are imported products (not locally produced); this seems to weaken the above authors’ explanation, at least in my opinion. 

9.Authors’ response: L324-328: The revisions requested by the reviewer were made by the authors in the article.

Comment on the Authors’ response: This part of the revised manuscript is identical with the corresponding part of the original version (lines 291-295) - no further clarification/rephrasing can be seen in the revised text. 

10.Authors’ response: L356-358: The revisions requested by the reviewer were made by the authors in the article.

Comment on the Authors’ response: Ok. Moreover, mentioning Table 4 (along with Table 5) might help further clarification, e.g. as follows: “Hooshfar et al. [27] reported that the HI value calculated for AFM1 in the analyzed baby foods was less than 1 (Table 5), as was also observed in our current study (Table 4), indicating that their consumption would not pose any health risk to babies. (Table 5)

11.Authors’ response: L370-371: The revisions requested by the reviewer were made by the authors in the article.

Comment on the Authors’ response: Ok.

12.Authors’ response: There is no difference. Infant formulas consumed by infants in Hatay are also sold and consumed by infants in other regions of the country.

Comment on the Authors’ response: Ok. On the other hand, if there is no difference and infant formulas consumed by infants in Hatay are also sold and consumed by infants in other regions of the country, the statement made in Conclusion (lines 400-403): “A notable strength of our study is the lack of previous research on aflatoxins and OTA in baby food samples from the study region, as well as the absence of studies examining dietary exposure and cancer risk related to baby food consumption in this area” seems less solid, at least in my opinion, and perhaps should be suitably modified.

13.Authors’ response: The literature review revealed that the number of samples collected in our study is higher than or comparable to those in other studies (see the link below). The authors deemed the number of collected samples sufficient for risk assessment analysis.

https://www.sciencedirect.com/science/article/abs/pii/S0956713512000412?via%3Dihub

https://pmc.ncbi.nlm.nih.gov/articles/PMC6468729/

Comment on the Authors’ response: Ok, I respect the authors’ opinion.

14.Authors’ response: No method was applied to the blank samples.

Comment on the Authors’ response: Ok. If not pre-analyzed with a different method, e.g. ELISA, I suppose that the blank samples were subjected to HPLC-FLD analysis both, before and after spiking, which is quite ok.

15.Authors’ response: The revisions requested by the reviewer were made by the authors in the article; L538: RFD: Reference dose (0.2 ng/kg b.w./day)

Comment on the Authors’ response: Ok.

16.Authors’ response: The data required for cancer risk assessment are provided here.

Comment on the Authors’ response: At least in my opinion, a better inter-connection of the data required for cancer risk assessment would improve quality of presentation.

Moreover, concerning “Quality of English language”:

English should be improved in a few parts of the text, e.g. lines 67-69 (“The stability of mycotoxins during processing and their contamination, particularly aflatoxins formed during post-harvest storage, in cereal-based infant formulas is inevitable); lines 71-73 (“However, since these toxins have different structures and require different methods for their detection, the use of similar techniques is not appropriate”); lines 324-328 (“It was stated that it was detected in the range of it is seen that the AFB1, AFM1 and OTA levels in the mentioned studies are higher than the levels detected in the current study, and it can be said that this is due to the fact that the contamination exposed during the production and storage stages of baby foods is higher in the samples analyzed in these studies”). Otherwise, English language is quite Ok.

Author Response

1. Comment on the Authors’ response: To help the authors keep the limit of 200 words, I would propose the following slight modification to the original abstract:

“In this study, it was aimed to determine the levels of aflatoxins and ochratoxin A (OTA) in baby food consumed in Hatay using fluorescence-detector HPLC (HPLC-FLD) and to reveal the health risks that through consumption of these foods may be occur in babies. To determine dietary intake and to reveal the health risk assessment, estimated daily intake (EDI) for all mycotoxins, margin of exposure (MOE) for aflatoxin B1 (AFB1), aflatoxin M1 (AFM1), and OTA, hazard index (HI) and consumption-related hepatocellular cancer risk for AFM1 were calculated. It was reported that (11.5%) and (8.2%) of analyzed samples exceeded the legal limit set for AFB1 and OTA, respectively. However, it was found that AFM1 concentrations in all samples did not exceed the legal limit. Based on the estimated consumption amounts of the baby foods, HI values calculated for AFM1 were below 1, and MOE values calculated for AFB1 and AFM1 were above 10.000, indicating that consumption of baby foods does not pose a risk regarding AFB1 and AFM1 for babies. However, it was determined in all other products, except for toddler formula that MOE values calculated for OTA were below 10.000, indicating that their consumption may pose serious health problems in babies”. Word count: 200

1. Authors’ response: The revisions requested by the reviewer were made by the authors in the article. Thank you for your help.

3. Comment on the Authors’ response: For better comprehension, I would propose rephrasing, e.g. as follows: “The rich content of baby foods in certain food ingredients that are necessary to support the physical and mental development of babies, along with their ever-increasing market-share, also make babies' exposure to toxic food-contaminants highly possible”.

3. Authors’ response: The revisions requested by the reviewer were made by the authors in the article. (L33-35). Thank you for your help.

8. Comments on the Authors’ response: 1. Please, add this information to “5.1. Sample Collection”. 2. According to the authors (Discussion, lines 234-237): “The lower AFB1 concentrations in the baby food samples analyzed in the current study compared to other studies may be attributed to the adherence to production and hygiene standards in feed and dairy products in Turkey, and the effective enforcement of established legal limits”. However, a substantial part of the baby foods analyzed in this study are imported products (not locally produced); this seems to weaken the above authors’ explanation, at least in my opinion.

8. The revisions requested by the reviewer were made by the authors in the article.

L412-413: The collected baby foods consists of both local (n=39) and imported (n=22) products.

Moreover, according to the authors, the fact that the number of local products is higher than imported products seems to support this sentence.

9. Comment on the Authors’ response: This part of the revised manuscript is identical with the corresponding part of the original version (lines 291-295) - no further clarification/rephrasing can be seen in the revised text.

9. Authors’ response: This section has been revised again in the article by authors.

L327-330: In these studies, the levels of AFB1, AFM1, and OTA detected in baby foods were observed to be higher than those detected in the current study. It can be said that this is due to the fact that baby foods produced in these countries are more contaminated with these toxins due to unsuitable production processes and storage conditions.

10. Comment on the Authors’ response: Ok. Moreover, mentioning Table 4 (along with Table 5) might help further clarification, e.g. as follows: “Hooshfar et al. [27] reported that the HI value calculated for AFM1 in the analyzed baby foods was less than 1 (Table 5), as was also observed in our current study (Table 4), indicating that their consumption would not pose any health risk to babies. (Table 5)

10. Authors’ response: This section has been revised again in the article by authors. (L358-361).

12.  Comment on the Authors’ response: Ok. On the other hand, if there is no difference and infant formulas consumed by infants in Hatay are also sold and consumed by infants in other regions of the country, the statement made in Conclusion (lines 400-403): “A notable strength of our study is the lack of previous research on aflatoxins and OTA in baby food samples from the study region, as well as the absence of studies examining dietary exposure and cancer risk related to baby food consumption in this area” seems less solid, at least in my opinion, and perhaps should be suitably modified.

12. Authors’ response: This section has been revised again in the article by authors.

L403-405: A notable strength of our study is the absence of a study revealing dietary risk assessment and cancer risk of aflatoxins and OTA exposure through baby food consumption in this region.

16. Comment on the Authors’ response: At least in my opinion, a better inter-connection of the data required for cancer risk assessment would improve quality of presentation.

Moreover, concerning “Quality of English language”:

English should be improved in a few parts of the text, e.g. lines 67-69 (“The stability of mycotoxins during processing and their contamination, particularly aflatoxins formed during post-harvest storage, in cereal-based infant formulas is inevitable); lines 71-73 (“However, since these toxins have different structures and require different methods for their detection, the use of similar techniques is not appropriate”); lines 324-328 (“It was stated that it was detected in the range of it is seen that the AFB1, AFM1 and OTA levels in the mentioned studies are higher than the levels detected in the current study, and it can be said that this is due to the fact that the contamination exposed during the production and storage stages of baby foods is higher in the samples analyzed in these studies”). Otherwise, English language is quite Ok.

16. Authors’ response: This section has been revised again in the article by authors. In addition to this, The information in this section includes the data used in the formula for calculating the cancer risk assessment. For this reason, we think that the data are interconnected. Language editing have been made in the specified sections.

L68-70: It is inevitable that aflatoxins, especially those formed during storage, are occur in cereal-based baby foods due to their stability during industrial processing.

L72-73: However, since these toxins have different structures, using similar techniques for their detection is not appropriate.

L327-330: In these studies, the levels of AFB1, AFM1, and OTA detected in baby foods were observed to be higher than those detected in the current study. It can be said that this is due to the fact that baby foods produced in these countries are more contaminated with these toxins due to unsuitable production processes and storage conditions.
